# Integrative Kinase Activity Profiling and Phosphoproteomics of *rd10* Mouse Retina during cGMP-Dependent Retinal Degeneration

**DOI:** 10.3390/ijms25063446

**Published:** 2024-03-19

**Authors:** Akanksha Roy, Jiaming Zhou, Merijn Nolet, Charlotte Welinder, Yu Zhu, François Paquet-Durand, John Groten, Tushar Tomar, Per Ekström

**Affiliations:** 1PamGene International B.V., 5200 BJ ‘s-Hertogenbosch, The Netherlandsjgroten@pamgene.com (J.G.); 2Division of Toxicology, Wageningen University and Research, 6708 WE Wageningen, The Netherlands; 3Ophthalmology, Department of Clinical Sciences Lund, Faculty of Medicine, Lund University, 22362 Lund, Sweden; 4Department of Science & Life, Fontys University of Applied Sciences, 5612 AR Eindhoven, The Netherlands; 5Mass Spectrometry, Department of Clinical Sciences Lund, Faculty of Medicine, Lund University, 22362 Lund, Sweden; 6Cell Death Mechanism Group, Institute for Ophthalmic Research, Eberhard-Karls-Universität Tübingen, 72072 Tübingen, Germany; yu.zhu@uni-tuebingen.de (Y.Z.); francois.paquet-durand@klinikum.uni-tuebingen.de (F.P.-D.); 7Lava Therapeutics N.V., 3584 CM Utrecht, The Netherlands

**Keywords:** *rd10*, PKG, CREB, CaMK, phosphoproteomics, kinome activity profiling

## Abstract

Inherited retinal degenerative diseases (IRDs) are a group of rare diseases that lead to a progressive loss of photoreceptor cells and, ultimately, blindness. The overactivation of cGMP-dependent protein kinase G (PKG), one of the key effectors of cGMP-signaling, was previously found to be involved in photoreceptor cell death and was studied in murine IRD models to elucidate the pathophysiology of retinal degeneration. However, PKG is a serine/threonine kinase (STK) with several hundred potential phosphorylation targets and, so far, little is known about the specificity of the target interaction and downstream effects of PKG activation. Here, we carried out both the kinome activity and phosphoproteomic profiling of organotypic retinal explant cultures derived from the *rd10* mouse model for IRD. After treating the explants with the PKG inhibitor CN03, an overall decrease in peptide phosphorylation was observed, with the most significant decrease occurring in seven peptides, including those from the known PKG substrate cyclic-AMP-response-element-binding CREB, but also Ca^2+^/calmodulin-dependent kinase (CaMK) peptides and TOP2A. The phosphoproteomic data, in turn, revealed proteins with decreased phosphorylation, as well as proteins with increased phosphorylation. The integration of both datasets identified common biological networks altered by PKG inhibition, which included kinases predominantly from the so-called AGC and CaMK families of kinases (e.g., PKG1, PKG2, PKA, CaMKs, RSKs, and AKTs). A pathway analysis confirmed the role of CREB, Calmodulin, mitogen-activated protein kinase (MAPK) and CREB modulation. Among the peptides and pathways that showed reduced phosphorylation activity, the substrates CREB, CaMK2, and CaMK4 were validated for their retinal localization and activity, using immunostaining and immunoblotting in the *rd10* retina. In summary, the integrative analysis of the kinome activity and phosphoproteomic data revealed both known and novel PKG substrates in a murine IRD model. This data establishes a basis for an improved understanding of the biological pathways involved in cGMP-mediated photoreceptor degeneration. Moreover, validated PKG targets like CREB and CaMKs merit exploration as novel (surrogate) biomarkers to determine the effects of a clinical PKG-targeted treatment for IRDs.

## 1. Introduction

Inherited retinal degeneration (IRD) diseases are rare neurodegenerative disorders characterized by progressive vision loss, primarily due to photoreceptor cell death. Mutations in over 300 genes have been identified as causal for IRD (https://sph.uth.edu/retnet; information retrieved 1 February 2024), making it imperative to develop gene and mutation-independent therapies that can serve larger patient populations. The accumulation of cyclic 3′,5′-guanosine monophosphate (cGMP) has emerged as a potential and crucial link in more than 20 IRD disease genes affecting at least 25% of IRD patients [1]. The overactivation of cGMP-dependent protein kinase G (PKG), one of the key effectors of cGMP-signaling, is likely to be involved in photoreceptor cell death [2,3,4]. The most frequently used murine model to study IRDs is the *rd1* mouse carrying a point mutation in exon 7 of the gene encoding for the beta subunit of phosphodiesterase 6 (*Pde6b*). The PDE6 enzyme is essential for the proper functioning of the photoreceptor since it hydrolyses cGMP in response to incoming light and, as such, governs phototransduction. The *rd1* mutation renders PDE6 non-functional, which results in an accumulation of cGMP and subsequent rod photoreceptor death [3,4]. In the *rd1* retina the cell death peaks at postnatal (P) day 12–P14 [4], while in the *rd10* mouse, another *Pde6b* mutation-based IRD model, with a missense mutation in exon 13 and an overall slower rate of photoreceptor degeneration, the rod cell death peaks around P18–P22. This means that the retinal structure and function remain intact over a longer period in *rd10* animals compared to *rd1*, and also that *rd10* photoreceptor degeneration does not overlap with the murine retina’s developmental cell death [5]. Together, this makes the *rd10* mouse a very attractive model for studying the pathophysiology behind such degeneration, as well as for the testing of protective treatments [3].

Much is already known about the structural and functional aspects of *rd10* degeneration, since these have previously been extensively characterized, including the activation of survival pathways [6,7,8,9]. Possible alterations to the *rd10* retinal proteome at pre-, peak-, and post-degenerative time points (P14, P21, and P28, respectively) have also been addressed [10]. A recent gene expression analysis of *rd10* at its peak cell death found cGMP-related genes to be affected, along with a dysregulation of transcription factors that regulate the stress response, apoptotic factor production, ion channel activity, optic nerve signal transduction, the metabolism, and intracellular homeostasis [11]. However, little is known about what happens in the *rd10* retina at the level of PKG-based phosphorylation, and an improved knowledge of these aspects would thus be most helpful for describing and understanding the degeneration machinery.

The fact that PKG is a serine/threonine kinase (STK) with several hundred potential phosphorylation targets likely provides complex cellular signaling during IRDs and makes it difficult to find targets or biomarkers relevant to photoreceptor cell death. To overcome this hurdle, here we combined kinase activity profiling and phosphoproteomics to identify potential and relevant PKG targets in the *rd10* retina. For this purpose, *rd10* organotypic retinal explants were treated with the PKG-specific inhibitor CN03 [3,12,13] and the effect was compared with untreated ones. Based on the profiling of kinome activity and phosphoproteomic data, several novel PKG substrates were identified and validated for their expression and location in both retinal tissues and explants. Overall, this study increases our knowledge of the cGMP/PKG-dependent photoreceptor degeneration mechanism and suggests relevant biomarkers for future PKG-based clinical therapies.

## 2. Results

### 2.1. PKG Inhibition Significantly Reduces Photoreceptor Cell Death in rd10 Retinal Explants

To link PKG inhibition with its protective effect on *rd10* photoreceptors, *rd10* retinal explants were treated with the well-established PKG inhibitor CN03, which binds to the cGMP-binding sites of PKG and competitively inhibits its activation [3,12]. The retinas of wild-type (WT) and *rd10* mice were explanted at P8, when photoreceptor degeneration had not yet started, and CN03 treatment was given to the *rd10* explants from P10 until P18, at which point the culture was terminated. Photoreceptor cell death was characterized by the TUNEL assay (Figure 1a), which labels dead cells. There were severalfold more TUNEL-positive cells in the *rd10* outer nuclear layer (ONL) compared to that of the WT explants, illustrating the rapid progression of the photoreceptor degeneration in this IRD model. Importantly, the CN03 treatment significantly reduced TUNEL-positive cells in the ONL to less than half of what was seen in untreated *rd10*, demonstrating the protective effect of PKG inhibition (Figure 1b).

### 2.2. PKG Inhibition Lowers Serine/Threonine Kinase Activity in rd10 Retinal Explants

As PKG inhibition significantly decreased photoreceptor cell death in the *rd10* retinal explants, the kinase activity profiles of the P18 *rd10* and P18 CN03-treated *rd10* retinas were determined on multiplex peptides microarrays, known as PamChip^®^ STK arrays. The overall kinome activity was decreased in the CN03-treated retinal explants (Figure 2a). Phosphorylation decreased in around 52% of the 142 peptides on the PamChip^®^ STK arrays. The phosphorylation of seven peptides, namely, CAC1C_1974_1986, ESR1_160_172, PLM_76_88, CREB1_126_138, PTK6_436_448, TOP2A_1463_1475, RBL2_655_667, CGHB_109_121, and STK_283_295, significantly (*p* < 0.05) decreased, whereas the phosphorylation of the peptide H2B1B_27_40 significantly increased with CN03 treatment in the *rd10* retinal explants (Figure 2b,c). Next, the changes in peptide phosphorylation between the two groups were linked to kinases potentially involved in that change by the “Upstream Kinase Analysis” tool of BioNavigator^®^ software (cf. Section 4.6, ‘Kinase activity measurements by kinome array’). The kinase score and kinase statistic were calculated as metrics to identify the kinases with potentially lowered activity. Most of the linked kinases came from the Ca^2+^/calmodulin-dependent kinase (CaMK), casein kinase 1 (CK1), and protein kinase A, G, and C (AGC) families, such as CaMK4, PKAα, CK1ε, Cyclin-dependent kinase-like 1 (CDKL1) protein kinase D1 (PKD1), PKG1, and PKG2, and were suggested to have reduced activity via the CN03 treatment (Table 1). The relative kinome activity profile of the retinal explants (*rd10* vs. *rd10* CN03) was represented on a phylogenetic tree of the protein kinase families with the color-coding of the branches according to the kinase statistic (Figure 2d). Here, the blue color represents decreased kinase activity in *rd10* CN03-treated retinal explants.

### 2.3. PKG Inhibition Alters the Phosphoproteome of rd10 Retinal Explants

To investigate how PKG also affected proteins not represented by the peptides on the chip, we undertook a global phosphoproteomic approach on retinal tissue in which the PKG activity had been inhibited. Proteins from explants either treated with CN03 or left untreated were extracted after explant homogenization and the precipitation of the soluble fraction, followed by phosphorylated peptide enrichment (based on an Fe-NTA Phosphopeptide Enrichment Kit; cf. Section 4.7 ‘Sample Preparation for MS’), and then analyzed with mass spectrometry (MS). In this way, 992 proteins with 597 phosphorylated peptides and 1002 phosphorylation sites were identified. In the further analysis a of at least five out of the nine samples (details in Section 4.8 ‘MS Acquisition and Analysis’), 711 phosphorylated sites in 263 proteins were finally compared from CN03-treated and untreated samples, and 94 phosphorylated sites of 54 proteins were identified as significantly different between the two groups (Figure 3a), with 28 and 66 phosphorylations of 21 and 36 proteins being decreased and increased, respectively (Figure 3b). Some of these proteins, and particularly those with reduced phosphorylation, could be potential cGMP-PKG-dependent substrates. Table 2 and Table 3 list the proteins whose phosphorylations on distinct peptides were either decreased or increased by the CN03 treatment, respectively. We note that two proteins, MAP1B and SRRM2, are present in the decreased as well as the increased phosphorylation categories, but these entries relate to different phosphorylation sites.

As several of the affected sites may also be linked to other kinases, we performed an additional upstream kinase analysis to reveal possible changes in kinome profiling. This identified 28 kinases with altered activities. Eight kinases came up as having proposed lower activities via the CN03 treatment, of which the mitogen-activated protein kinase (MAPK) family took up the majority, including MAPK1, MAPK3, and MAPK14, whereas Ca^2+^/calmodulin-dependent protein kinase II alpha (CaMK2a), MAPK14, and tyrosine-protein kinase ZAP-70 (ZAP70) decreased the most. In contrast, we observed 20 kinases with higher activities from varied kinase groups, among which ribosomal protein S6 kinase beta-1 (RPS6KB1 or p70S6 kinase), cyclin-dependent kinase 2 (CDK2), and FYN proto-oncogene (FYN) increased the most (Figure 3c).

### 2.4. Potential Biological Pathways Involved in Retinal Degeneration

Next, we integrated kinase activity profiling data and phosphoproteomic data to identify possible common networks of kinases and biological pathways. The predicted kinase list from both datasets (Table 2 and Appendix A) were entered into the STRING database [14] to find common networks and their connections (Figure 4).

The highest-ranking network of kinases common to both types of generated data were PKG1, PKG2, CaMK4, CaMK2α, MAPKs, AKTs, and RSKs, among others. To link this kinase-based network to a biological context, we performed a pathway analysis using the Reactome database; the major associated pathways with potentially altered activity after CN03 treatment in *rd10* explants were “Intracellular signaling by second messengers”, “Calcium induced signaling”, “Calmodulin induced events”, “MAPK targets”, “CREB phosphorylation”, and “RAS activation”. Notably, we also observed pathways like “Cell cycle activation” and “Suppression of apoptosis” among the top-predicted pathways, which supports the overall protective effect of PKG inhibitors in cGMP-mediated retinal degeneration.

### 2.5. Confirmation of CaMK and CREB Phosphorylation in the Retina

Based on the kinome profiling and phosphoproteomic data, as well as the pathway analysis, we tested key proteins that relate to the differential phosphorylation in the CN03-treated *rd10* retina compared to the untreated situation. For this, we selected phosphorylated CREB (pCREB), plus CaMK2 and CaMK4 together with their phosphorylated variants (pCaMK2, pCaMK4). The rationale was that CREB is regarded as a substrate of cGMP-PKG [15], as well as that it has been shown to connect with retinal degeneration [16]. Together with the fact that the CREB peptide on the chip (see above) showed reduced phosphorylation in the CN03-treated situation (Figure 2), this suggested that an analysis of pCREB would be highly relevant. Moreover, the activities of CaMK2 and CaMK4 were indicated to be affected by the CN03 treatment by the phosphoproteomics-based upstream kinase analysis and the chip-based kinase profiling, respectively (Figure 2d and Figure 3c). CaMK2 has previously been studied in retinal degeneration, where its activity is altered [17], but the possibility that it is a substrate for PKG was not addressed. All in all, this prompted us to look closer at this aspect for both CaMK2 and CaMK4.

It is crucial to understand where the suggested protein targets of PKG are localized in the retina, since a localization outside the actual photoreceptors would be unlikely to have an involvement in degeneration. Immunostaining was thus performed for retinal sections derived from WT and *rd10* P18 mice (Figure 5). CaMK2 was found to be expressed in the outer plexiform layer (OPL), inner nuclear layer (INL), inner plexiform layer (IPL), and ganglion cell layer (GCL) for both the WT and *rd10*, with a perhaps slightly weaker signal intensity in the *rd10* INL. Phosphorylated CaMK2 (pCaMK2, at threonine 287) was located in INL and GCL in the WT and *rd10*, with a clearly stronger signal in the *rd10*. The signal of phosphorylated CREB (pCREB, at serine 133) was predominantly seen in the OPL, INL, IPL, and GCL and more so in the *rd10* than WT. In contrast, CaMK4 seemed to be more abundant in the OPL, INL, IPL and GCL of the WT rather than *rd10*. Phosphorylated CaMK4 (pCaMK4, at threonine 196/200) showed a higher signal intensity in the ONL of the WT compared to *rd10*. Since an abnormally high level of cGMP, as well as its dependent PKG activity, are mainly observed in photoreceptors, a focused comparison of the fluorescence signal of these proteins was performed within the ONL between the two strains. As seen in the bar diagram of Figure 5, there was no ONL difference for CaMK2, while the signals of pCaMK2 and pCREB were stronger in *rd10* than in the WT. By contrast, a higher signal for CaMK4 and its phosphorylation was observed in the WT ONL. Therefore, the localization of the selected PKG targets was confirmed in the WT and *rd10* retinal tissues. The higher ONL signal intensity of pCREB and pCaMK2 in the *rd10* situation and the lower signal for pCaMK4 demonstrated the differential effects of the *rd10* degeneration on the selected proteins. It should be noted that antibodies may bind to proteins in an unspecific manner, even though our Western blotting does not indicate any such cross-reactivity (see next paragraph and Figure 6).

Moreover, the immunostaining for pCREB and pCaMK4 gave opposite ONL relationships in the WT–*rd10* comparison (Figure 5), suggesting that the antibody signals do not simply represent a general, indiscriminate reaction to any retinal protein. A similar relationship was observed in the Western blot results, where the treatment of *rd10* explants with CN03 gave opposite outcomes for pCREB and pCaMK4 (Figure 6). Such differentiating results again make general and unspecific binding unlikely.

To further analyze and quantify the activation of the selected targets CaMK2, CaMK4, and CREB, we performed immunoblotting of in vivo retinas and/or whole retinal explant lysates using antibodies for the activated variants, i.e., pCaMK2, pCaMK4, and pCREB (Figure 6). This allowed us to make strict WT–*rd10* comparisons (Figure 6a,b), as well as see what the CN03 treatment did to the activation state in the *rd10* explants (Figure 6c,d).

For the WT–*rd10* comparisons, there was an increased presence of both pCREB and pCaMK2 in the *rd10* case, where the difference for pCaMK2 was statistically significant, whereas pCaMK4, by contrast, was significantly higher in the WT. The WT vs. *rd10* comparison, therefore, replicated the corresponding immunostainings for pCaMK2 and pCaMK4 in the ONL, where pCaMK2 was lower, and pCaMK4 was higher, in the WT (cf. Figure 5). In the CN03-treated vs. untreated comparisons, the CN03 treatment decreased the pCREB level and increased that of pCaMK4. The CN03 treatment thus appeared to have normalized the pCaMK4 levels in the *rd10* retina, since the treated explants displayed an increased presence of the pCaMK4 signal, i.e., being more like the WT, as in the WT vs. *rd10* comparison (Figure 6d). The *rd10* pCREB level was, in both the immunostaining and immunoblotting analyses, numerically higher than that in the WT, although this did not reach statistical significance. Also, here the CN03 treatment acted to normalize the situation, since, in the immunoblotting, the pCREB level in the treated specimens was clearly lower than in the untreated *rd10* (Figure 6d), i.e., it was brought closer to the WT level.

## 3. Discussion

There are animal models, including mouse-based ones, that faithfully resemble human retinal degenerations to the extent that this has contributed to now-established therapies, like that for the rpe65 gene in Leber’s congenital amaurosis [18,19]. While this must not be taken to mean that results from any animal model can be extrapolated to the clinical level, it is clear that mutations in the human gene for PDE6b cause photoreceptor degeneration, similar to what is seen in the *rd10* model used here [20,21]. Together with the fact that the peptide sequences used in the PamGene chip are based on our knowledge of the corresponding human proteins (see Section 4.6. ‘Kinase Activity Measurements by Kinome Array’), this would act to shorten the distance between our use of this model and the understanding of patients.

A number of IRD-causing gene mutations converge on a cGMP-driven PKG-signaling route, suggesting that targeting PKGs could provide a new, common treatment that may benefit many IRD patients [1,4]. In line with this, PKG inhibition has afforded photoreceptor protection in several in vivo and ex vivo IRD models [3,4]. Here, our integrative kinome activity and phosphoproteomic study showed that the destructive effects of PKG-overactivation might be linked to several downstream PKG targets, including CREB and the Ca^2+^/calmodulin-dependent protein kinases CaMK2 and CaMK4. Our results, therefore, provide new insights towards the understanding cGMP-mediated retinal degenerative diseases and may be instrumental for the design of new therapies as well as biomarkers to assess therapeutic efficacy.

One should be reminded, though, that these observations, as such, do not provide conclusive evidence of a mechanistic involvement of any of the substrates, be they known or novel. For instance, some of the observed alterations could be epiphenomena triggered by the PKG inhibition but unrelated to the degeneration. Moreover, to reveal a direct mechanistic involvement of the PKG substrates would require further investigations well beyond the scope of the present study.

### Identifying Protein Targets for PKG Phosphorylation

Previous studies have identified possible PKG targets such as the VASP (vasodilator-stimulated phosphoprotein; [3]). Besides such individual target-specific approaches, more holistic tactics, like the MS-centric proteome analysis or kinome substrate-specific peptide microarray, have each, on their own, been implemented to identify novel kinases and other enzyme targets in the retina [22,23]. Moreover, with the help of the proteomic and kinomic approaches, several new cGMP-interacting proteins have been found, including CaMK2a, MAPK1/3, and Glycogen synthase kinase 3β [24], as well as potential downstream signaling nodes such as the calcium/potassium channel and VASP axis in the *rd1* retina [13]. However, a careful integration of phosphoproteomic and kinomic data generated from the same experimental study would provide a better understanding of the molecular mediators at the protein level, as well as of the phosphorylation levels that regulate the underlying cGMP-mediated retinal degeneration pathways. In addition, any in-this-way identified kinases or proteins may serve as potential biomarkers for PKG-targeted therapy and its response.

Our multiplex kinase activity profiling identified several peptides with decreased phosphorylation after PKG inhibition. Of these peptides, CAC1C_1974_1986, CREB1_126_138, PLM_76_88, PTK6_436_448, RBL2_655_667, STK6_283_295, and TOP2A_1463_1475 have been reported as PKG1 and/or PKG2 substrates [22]. Interestingly, H2B1B_27_40, the only peptide that showed increased phosphorylation with CN03 treatment, was also identified as a potential good PKG2 substrate in the same study. The decrease in peptide phosphorylation due to CN03 treatment was linked to potential kinases such as the Ca^2+^-calmodulin activated CaMK2 and CaMK4 [25], as well as PKGs and PKA. Of these, PKG, PKA, and CaMK2 have themselves been determined to be engaged, or have an increased likelihood for activation (higher cAMP), in *rd* retinas and have all been identified as potential cGMP-interacting proteins through LCMS-based proteomic analysis [17,23,26]. Both CaMK2a and CaMK4 were, furthermore, amongst the kinases predicted to have lower activities in *rd1* retinas with short-term PKG inhibition (see supplementary material of [23]). It is thus possible that the proteins corresponding to the mentioned peptides are important PKG targets during the *rd10* photoreceptor degeneration. Similar results on the role of CaMK, PKA, and CREB have been reported recently using the *rd1* explant model treated with CN03 [13]. Interestingly, in this model, significant treatment effects were seen in PKG substrates that were related to the potassium channels’ K_v_1 family (KCNA3 and KNCNA6) and their presence in relevant parts of the retina was confirmed by immunostaining. This CN03 effect was not so obvious in the *rd10* model, which merits further examinations employing straightforward head-to-head comparisons of PKG targets.

The multiplex approach makes use of 3D microarray chips with several defined substrate peptides to known STKs. To allow for the detection of other, not yet defined substrates, in addition, we performed a phosphoproteomic analysis. Since some bias might be introduced during such work, it is pertinent to reflect on this. The manufacturers of the Fe-NTA Enrichment Kit (see Section 4.8 ‘MS Acquisition and Analysis’) recommend the use of at least 500 µg of protein digest for optimal results. Using lower amounts of a protein digest sample is possible, but the phosphopeptide yield will be significantly lower with potentially more non-phosphopeptides present after enrichment, calling for extra care.

Because we had limited quantities of the starting material, we kept the amount for all samples the same (i.e., 150 µg of the starting material for protein digestion), and also kept all volumes for all the samples during reduction and alkylation the same (i.e., 100 µL). The proteins were precipitated with 900 µL of ice-cold ethanol. Before phospho-enrichment, we resuspended all the samples in 200 µL Binding/Wash buffer per the manufacturer’s protocol. This, together with the fact that the outcomes had similarities with those from phosphoproteomics and other studies on a different cGMP-related model, *rd1* (see below; [13,17,23]), suggests that the results can be regarded as reliable.

Here, we detected 28 sites (on 21 proteins) with reduced phosphorylation after PKG inhibition. According to the upstream kinase analysis, some of these may link to the reduced activities of kinases from the MAPK family, including ERK1, ERK2, and p38 (Figure 3c). The MAPK family plays a key role in cell proliferation, differentiation, and stress responses and the ERK kinases are involved in signaling cascades and the transmission of extracellular signals to intracellular targets [27]. PKG mediated-signaling has been determined to be necessary, e.g., for the prolonged activation of MAPK in invertebrate associative memory [28], and our finding of the PKG inhibitor-mediated reduction in ERKs and p38 MAPK therefore indirectly indicated that cGMP-PKG upregulated the activities of these kinases, in turn, leading to detrimental effects during degeneration. This would be consistent with a recent report that pharmacological ERK1/2 inhibition had protective effects against photoreceptor degeneration [29]. A reduction in the activities of retinal ERK kinases via long-term PKG inhibition was also suggested by a very recent study on the similar *rd1* model [24], which underlines the current finding.

Remarkably, our results regarding CaMK2a in *rd10* are very compatible with previous work on the likewise cGMP-connected *rd1* model, with high PKG activity [3], that was obtained by independent methods involving 2D-gel electrophoresis-based proteomics, and which concluded there was increased CaMK2a activity in *rd1* compared to the WT [17]. Our kinome profiling suggested reduced CaMK2a activity after PKG inhibition in *rd10*, and consequently, given the similarities of the models, this would fit with PKG being responsible for the increased CaMK2a activity in *rd1* [17]. Moreover, the previous work showed that the photoreceptor-specific CaMK2a substrate phosducin showed increased phosphorylation in *rd1*, whereas the present phosphoproteomic approach correspondingly showed that PKG inhibition decreased phosducin phosphorylation. Time event studies to describe the (de-)phosphorylation of phosducin are therefore needed to shed more light on the interplay with CaMK targets and their phenotypic readouts in mouse models. However, already these studies, together with the suggested reduction of CaMK2a activity after PKG inhibition in *rd1* [23], give a strong argument for a CaMK2a involvement downstream of PKG in photoreceptor degeneration. This is well-matched with the knowledge that PKG can activate calmodulin and CaMK2-dependent intracellular mechanisms [30]. The role of calmodulin and CaMKs in PKG-related photoreceptor degeneration was indeed further suggested by the identification of “Calmodulin induced events” as one of the major biological pathways potentially affected by CN03 treatment. Our work, therefore, furthers CaMK2 and CaMK4 as potential players in degeneration, although the picture is quite complicated. The present results are consistent with the higher CaMK2 activity in *rd10* (higher levels of cCaMK2a than in the WT, and reduced activity in *rd10* after PKG inhibition, as per the upstream kinase analysis). Yet, the CaMK4 data, in contrast, point to lower CaMK4 activity in this model (lower levels of pCaMK4 than in the WT, and also increased levels when *rd10* is treated with CN03). Both kinases have been connected with neuronal survival or protection, since CaMK2 has been reported to protect retinal ganglion cells in glaucoma [31], while CaMK4 promotes neural survival [32] and is reduced in neurodegenerative diseases, such as amyotrophic lateral sclerosis [33]. One possible scenario could therefore be that the high *rd10* CaMK2 activity represents an ongoing protective response, initiated by the PKG activity in parallel with degenerative events, whereas a possible protective CaMK4 activity, by contrast, is prohibited by the very same PKG actions.

In addition to proteins with reduced phosphorylation, we identified 66 sites (on 36 proteins) that showed increased phosphorylation when PKG was inhibited, suggesting that the cGMP-PKG system could also negatively regulate other kinases. Kinases with higher activities under the CN03 treatment, as suggested by the kinome profiling, included three members of the ribosomal S6 kinase (RSK) family, ribosomal protein S6 kinase B1 (RPS6KB1, highest activity), ribosomal protein S6 kinase B2 (RPS6KB2) and ribosomal protein S6 kinase A1 (RPS6KA1), of which, the latter promotes cellular survival in a model of BAD-modulated cell death [34]. Also, AKT serine/threonine kinase 1 (AKT1) [35] and glycogen synthase kinase-3β (GSK-3β) [36], with known survival effects, were indicated to have higher activities after PKG inhibition. It is noteworthy that GSK-3β was recently shown to be a potential cGMP-interacting protein in the proteomic analysis of the retina in the *rd10* model [24], and future investigations may reveal whether such interactions entail a PKG-mediated regulation of GSK-3β activity. One general interpretation of the kinase activation data could be that the cGMP-dependent photoreceptor death involves the inhibition of kinases with an anti-cell death function during retinal degeneration.

Based on the integration of results from both the kinase activity profiling and the phosphoproteomics, CREB, CaMK2, and CaMK4, and/or their phosphorylated variants, were selected as targets for further analyses, and the outcome for the CaMKs has been discussed above. CREB is a known PKG substrate [22] that stimulates transcription on binding to the DNA cAMP response element and is involved in neuronal survival [37]. In our previous study, CREB showed increased phosphorylation in *rd1* retinal explants and was confirmed as a potential PKG target [22], and has also previously been connected with retinal degeneration in the *rd1* model [16]. According to the in vitro studies here, the phosphorylation of CREB was decreased via CN03 treatment, further supporting that this step is PKG-mediated. CREB is a target for the kinase CaMK2 [38], as well as CaMK4 [39]. Since CaMKs and CREB were also identified as potential proteins with reduced phosphorylation after CN03 treatment in the *rd1* model [13], PKG-mediated CaMKs-CREB signaling appears to be a promising target to develop potential biomarkers for retinal disease progression and therapy response.

In summary, the effect of PKG inhibitor treatments showed a significant overlap between *rd10* and previously reported data in *rd1*. The major common peptides with significantly decreased phosphorylation were CREB and TOP2A, whereas the kinases potentially involved in differential phosphorylation were PKG1, PKG2, and CaMK4. It seems, though, that the CN03 treatment of *rd10* retinas resulted in a lower number of affected peptides and phosphoproteomic-identified proteins when compared to a similar treatment in *rd1* [13,23]. Such differences could be attributed to the distinctive rates of photoreceptor degeneration in *rd1* and *rd10*, as well as the fact that the time point chosen for CN03 treatment was different between the models. There is also some residual PDE6 activity still left in *rd10* at day 18 [40], which may have contributed to this. A dedicated comparison of PKG targets in both *rd1* and *rd10* retinas to investigate the expression of these potential PKG targets with time is therefore advised, and should be addressed in future studies assessing retinal degeneration.

## 4. Materials and Methods

### 4.1. Organotypic Retinal Explant Cultures

The following two mouse lines were used in this study: C57BL/6J *rd10*/*rd10* (*rd10*, RRID:MGI:3581193, The Jackson Laboratory, Bar Harbor, ME, USA) and a control C3H wild-type (WT, [41]). Both lines were housed under regular white cyclic lighting, had free access to food and water, and were used irrespective of sex. All the procedures were performed according to the issued local animal ethics committees (permits 02124-2020, AK02-19M) and the ARVO statement for the use of animals in ophthalmic and visual research. All efforts were made to minimize the number of animals used and their suffering. The study was not preregistered.

Retinas from P8 *rd10* and WT mice were used to generate explants following the standard protocol as previously described [3]. Mice were euthanized and their eyes were rapidly enucleated and incubated in an R16 medium (07491252A, Gibco, Waltham, MA, USA) and treated for 15 min with 0.12% proteinase K (21935025, ICN Biomedicals Inc., Irvine, CA, USA), the activity of which was subsequently blocked by 10% fetal bovine serum (F7524, Sigma-Aldrich, Darmstadt, Germany) followed by rinsing in an R16 medium. In a sterile environment under a laminar-flow hood, the retina with the retinal pigment epithelium (RPE) attached was separated from the eyes and the anterior segment, lens, vitreous, sclera, and choroids were removed. The retina was then incised to give a four-leaf clover shape and transferred to a culture membrane insert (3412; Corning Life Sciences, Corning, NY, USA), with the RPE directly facing the insert membrane. The inserts with the explants were then put into six-well culture plates (83.3920, Sarstedt, Nümbrecht, Germany) with 1.5 mL of serum-free R16 medium with supplements added to each well [3], incubated at 37 °C with a CO_2_ level of 5%, and with the medium replaced every second day. Retinas were selected randomly for either treatment or control. For the *rd10* retinas, the first 2 days in culture were without any treatment, and after this, i.e., at a time that equals P10, the cultures were exposed to 50 μM of Rp-8-Br-PET-cGMPS (also known as CN03 [3,4], PKG inhibitor; Cat. No.: P 007, Biolog, Bremen, Germany), respectively, with their corresponding untreated controls receiving an equal amount of solvent (water). The end point of the whole culturing procedure was after another eight days of culture, i.e., equivalent to P18. The same paradigm (P8 + 10 days) was applied to the WT without any treatments. All retinal explant samples were collected for kinome activity microarray measurements or phosphoproteomic based analyses, or fixed, sectioned, and used for microscopy-based analyses.

### 4.2. Cryosection and Immunostaining

Retinal tissues from *rd10* and WT in vivo at P18, as well as the P18 cultured explants, were treated with 4% formaldehyde for 2 h, washed 4 × 15 min in phosphate-buffered saline (PBS), cryoprotected in PBS + 10% sucrose overnight at 4 °C and subsequently with PBS + 25% sucrose for 2 h. After embedding, 12 μm thick retinal cross-sections were cut and collected from an HM560 cryotome (Microm, Walldorf, Germany). The sections were stored at −20 °C for later usage. Cryosections were used for immunostaining as described before [4]. Briefly, the cryosections were dried at room temperature for 15 min and rehydrated in PBS. They were then blocked with 1% BSA + 0.25% Triton X100 + 5% goat serum in PBS at room temperature for 45 min. Primary antibodies (pCREB: #MA5-11192; CaMK2: #PA5-82911; pCaMK2: #PA5-37833; CaMK4: # PA1-542; pCaMK4 all from ThermoFisher, Uppsala, Sweden; and #SAB4504122, Sigma-Aldrich) were diluted with 1% BSA and 0.25% Triton X100 in PBS (PTX) and incubated at 4 °C overnight at a 1:200 dilution; a no-primary-antibody control ran in parallel. After the incubation, the sections were washed 3 × 5 min each in PTX and incubated with a goat anti-rabbit IgG (H+L) cross-adsorbed secondary antibody with Alexa Fluor 594 (#A11037, ThermoFisher) at a 1:400 dilution in PTX. After 3 × 5 min PBS washes, the sections were mounted with Vectashield DAPI (Vector, Burlingame, CA, USA).

### 4.3. TUNEL Assay

To evaluate the neuroprotective effects of CN03 on photoreceptor death, a fluorescent terminal deoxynucleotidyl transferase dUTP nick end labeling (TUNEL) assay (#11687495910, Roche Diagnostics, Basel, Switzerland) was performed according to the manufacturer’s instructions and used on cryosections generated from the *rd10* and WT retinal explant cultures.

### 4.4. Microscopy and Image Processing

A Zeiss Imager Z1 Apotome Microscope (Zeiss, Oberkichen, Germany), with a Zeiss Axiocam digital camera was used for the microscopy-based observations. In further processing, image generation and contrast enhancement were performed identically for all images via the use of ZEN2 software (blue edition; Zeiss, Oberkichen, Germany). The immunostaining and TUNEL were analyzed for staining differences via three sections from three-to-six animals for each condition, after which the fluorescent intensities of positive cells randomly distributed within in the selected area of interest (outer nuclear layer, ONL, i.e., the photoreceptor layer) was assessed. Fluorescence intensity was captured and analyzed by ImageJ software (version 1.53a, NIH, MD, USA), where the freehand selection function was used to outline the ONL, with the fluorescence intensity calculated with the measure function. The values of all sections from the same animal were averaged before further analysis.

### 4.5. Retinal Explant Lysis

Mammalian protein extraction reagent (M-PERTM), HaltTM protease and phosphatase inhibitor cocktails, and the Coomassie Plus (Bradford Assay) kit were all purchased from Thermo Fischer Scientific. Whole retinal explant lysates (*rd10* NT *n* = 10, *rd10* CN03 *n* = 10) were prepared in a Lysis Buffer (M-PER with 1:100 Protease and Phosphatase Inhibitor cocktails), such that the samples were lysed for 30 min on ice followed by centrifugation (16,000× *g* for 15 min at 4 °C). The supernatant was subsequently divided into aliquots, snap frozen and stored a −80 °C until further processing. Protein quantification was performed with a Bradford Assay as per the manufacturer’s instructions [42].

### 4.6. Kinase Activity Measurements by Kinome Array

Kinase Activity profiling was performed on Serine/Threonine Kinase (STK) PamChips^®^, where each chip comprises four arrays with 142 serine/threonine-containing immobilized-peptides which are derived from human phosphoproteome, according to the instructions of the manufacturer (PamGene International B.V., ‘s-hertogenbosch, The Netherlands). Briefly, an assay mix was prepared with 0.25 µg of protein lysate, a protein kinase buffer (proprietary, PamGene), 0.01% BSA, STK primary antibody mix (proprietary, PamGene), and 400 µM of ATP. Firstly, the PamChips^®^ were placed in the PamStation12^®^ system and blocked with 2% BSA. Subsequently, an assay mix containing active kinases was added and pumped back and forth through PamChip^®^ wells to facilitate interaction between the active kinases and the 142 immobilized consensus phospho-peptide sequences. The presence of peptides phosphorylated by kinases present in the retinal lysate and their extent of phosphorylation were assessed with an FITC-conjugated secondary antibody targeting towards the primary STK antibody cocktail [43,44]. The images of the arrays were recorded at multiple exposure times and the signal intensity of each peptide was quantified by BioNavigator^®^ software version 6.3.67.0 (PamGene International B.V., ‘s-Hertogenbosch, North Brabant, The Netherlands). The signal intensity at multiple time points was combined to a single value and log2 transformed. The overall differences in the *rd10*-untreated (*rd10* NT) and CN03-treated (*rd10* CN03) samples’ STK peptide phosphorylation profiles were visualized as violin plots (GraphPad Prism version 9.2.0). Significant differences (*p* < 0.05) in phosphorylation intensity between the two groups were determined via a Paired *t*-test and results were represented as volcano plots (GraphPad Prism version 9.2.0). The kinases that might be responsible for differences in the peptide phosphorylation between the treated and untreated retinal samples were ascertained by the STK Upstream Kinase Analysis tool of BioNavigator^®^. This software combines the information of kinase interaction with phosphorylation sites from databases such as HPRD, PhosphoELM, PhosphositePLUS, Reactome, UniProt [43]. The highest-ranking predicted kinases based on their kinase score (significance and specificity score) were represented on a phylogenetic tree of the human protein kinase family (Coral, http://phanstiel-lab.med.unc.edu/CORAL/; accessed 3 November 2021) [45].

### 4.7. Sample Preparation for MS

Each retinal explant was homogenized separately in a buffer (50 mM of Tris-HCl, 50 mM of NaCl, 1 mM of EDTA, 5 mM of NaH2PO4, 1 mM of DL-Dithiothreitol [DTT]), supplemented with phosphatase inhibitors (Lot No, 33041800, Roche, Basel, Switzerland, 1 tablet per 10 mL of buffer) using a homogenizer (Knotes Glass Company, Vineland, NJ, USA). The homogenate was then centrifuged at 10,000× *g* for 5 min at 4 °C, after which the soluble fraction was collected, with the concentration measured using a Bio-Rad Protein Reagent Assay Kit (Cat. No.: #5000113, #5000114, #5000115, Bio-Rad, Hercules, CA, USA).

For each separated sample, 150 µg of proteins were reduced with DTT to a final concentration of 10 mM and heated at 56 °C for 30 min followed by alkylation with iodoacetamide for 30 min at room temperature in the dark, to a final concentration of 20 mM. Subsequently, samples were precipitated with ice-cold ethanol overnight at −20 °C, followed by centrifugation at 14,000× *g* for 10 min, after which the pellets were resuspended in 100 mM of ammonium bicarbonate and sonicated for 20 cycles of 15 s on, 15 s off, using a Bioruptor (Diagenode, Denville, NJ, USA). A digestion was then performed by adding trypsin (Sequencing Grade Modified Trypsin, Part No. V511A, Promega, Madison, WI, USA) at a ratio of 1:50 to the samples and incubated overnight at 37 °C, after which it was stopped by 5 µL of 10% trifluoroacetic acid (TFA). The Pierce High-Select Fe-NTA phosphopeptide Enrichment Kit (Cat. No.: A32992; Thermo Fischer Scientific, Waltham, MA, USA) was used to enrich phosphopeptides according to the manufacturer’s protocol. The phosphopeptides were run to dryness in a Speed Vac and resolved in 2% acetonitrile (ACN) and 0.1% TFA to a peptide concentration of 0.25 µg/µL.

### 4.8. MS Acquisition and Analysis

LC-MS detection was performed on a Tribrid mass spectrometer Fusion (Thermo Fischer Scientific) according to Rasmussen et al.’s 2021 study [24]. Briefly, for each sample, 1 µg of the peptides was injected into the LC-MS. Peptides were concentrated on an Acclaim PepMap 100 C18 precolumn (75 μm × 2 cm, Cat. No.: 164941, Thermo Fischer Scientific) for subsequent separation on an Acclaim PepMap RSLC column (75 μm × 25 cm, C18, 2 μm, 100 Å, nanoViper, Cat. No.: 164941, Thermo Fischer Scientific) at a temperature of 45 °C and a flow rate of 300 nL/min. Solvent A (0.1% formic acid in water) and solvent B (0.1% formic acid in can) were used to create a nonlinear gradient for the elution of the peptides. The gradient was constructed such that the percentage of solvent B was maintained at 3% for 3 min, increased from 3% to 30% for 90 min, increased to 60% for 15 min, and then increased to 90% for 5 min to be kept at 90% for another 7 min to wash the column.

The Orbitrap Fusion was operated in positive data-dependent acquisition (DDA) mode. The peptides were introduced into the LC-MS via a stainless steel Nano-bore emitter (OD 150 µm, ID 30 µm) with the spray voltage set to 2 kV and the capillary temperature to 275 °C. Full MS survey scans from *m*/*z* 350–1350 with a resolution of 120,000 were performed in the Orbitrap detector. The automatic gain control (AGC) target was set to 4 × 105 with an injection time of 50 ms. The most intense ions (up to 20) with charge states 2–5 from the full scan MS were selected for fragmentation in the Orbitrap. The MS2 precursors were isolated with a quadrupole mass filter set to a width of 1.2 *m*/*z*. Precursors were fragmented via high-energy collision dissociation (HCD) at a normalized collision energy (NCE) of 30%. The resolution was fixed at 30,000 and for the MS/MS scans, the values for the AGC kinase group and injection time were 5 × 10^4^ and 54 ms, respectively. The duration of the dynamic exclusion was set to 45s and the mass tolerance window was 10 ppm.

The raw DDA data were analyzed with Proteome Discoverer™ Software (Version 2.5, Thermo Fisher Scientific). Peptides were identified using both SEQUEST HT [46] and Mascot [47] against the UniProtKB mouse database (UP000000589 plus isoforms). The search was performed with the following parameters applied: static modification, cysteine carbamidomethylation and dynamic modifications, and N-terminal acetylation. The phosphorylation (S, T, Y) was set as variable for the phosphopeptide analysis. Precursor tolerance was set to 10 ppm and fragment tolerance was set to 0.05 ppm. Up to two missed cleavages were allowed and Percolator was used for peptide validation at a maximum q-value of 0.05. Extracted peptides were used to identify and quantify them by label-free relative quantification. The extracted chromatographic intensities were used to compare peptide abundances across samples.

The MS results were processed via Perseus software (version 1.6.0.7) [48]. The protein intensities were log2 transformed. The selection criteria were defined as follows: among the 5 and 4 replicates of the two conditions, respectively (5 from *rd10* with CN03 treatment and 4 from untreated *rd10*, in total 9 samples), only peptides which presented with values in more than 50% of all 9 samples (meaning that a missing value was allowed in a maximum of 4 of the 9 smaples) would be selected, and the missing values were replaced from a normal distribution that was performed through data imputation using the following settings: width 0.3 and downshift 0. The further bioinformatics analysis of these processed data was done via the web-based tool Phosphomatics (https://phosphomatics.com/; accessed 3 November 2021), ([49]; UniProt: a worldwide hub of protein knowledge). A two-sample Student’s *t*-test (two-tailed) was performed to compare phosphorylated site levels between the *rd10* explants with PKG inhibition and their counterparts. A *p*-value of 0.05 was defined as the cut-off.

### 4.9. Integrative Kinome Network and Pathway Analysis

To perform an integrative network analysis, we used the String database (https://string-db.org/; accessed 3 November 2021), which is a comprehensive resource for curated organism-wide protein associations and here used to integrate known and predicted associations between proteins, including both the physical interactions as well as functional associations [50]. We used differentially predicted upstream kinases from kinome activity microarray data (*n* = 17 from Table 1) and MS-centric phosphoproteome data (*n* = 28 from Appendix A) as inputs for the String DB network analysis. For the network analysis, only a medium confidence (0.04 for an interaction score) and a Markov cluster algorithm were applied with default settings. The biological pathways that might be affected between untreated and CN03-treated *rd10* samples were determined via a Reactome Pathways analysis (https://reactome.org/, accessed 3 November 2021 [51]), and here the same differentially predicted upstream kinases as that of the network analysis were used as an input list. Visualization was performed in GraphPad Prism (version 9.2.0) using known matrices from the analysis.

### 4.10. Western Blot

An amount of 20 μg of retinal explant lysate per sample per lane were loaded and separated on 4–20% Mini-PROTEAN^®^ TGX Stain-Free™ Protein Gels (Bio-Rad, Hercules, CA, USA, Cat. #4568096) at 150 V and 40 A for 50–60 min in a 1× Tris/Glycine/SDS (TGS) buffer (Bio-Rad, #1610732). All proteins were transferred to a Trans-Blot Turbo Mini 0.2 µm PVDF Membrane (Bio-Rad, Cat. #1704156) using the Trans-Blot Turbo Transfer System (Bio-Rad). The Trans-Blot system was set to run for 7 min at 1.3 A and 25 V per blot. For staining, the membrane was blocked for 1 h at room temperature (RT) with 5% low-fat milk (ELK, Campina) in PBST (Phosphate-buffered saline with 0.1% Tween-20) or with 5% BSA in PBST for phospho-proteins. Before adding the primary antibody, the membrane was washed 3× times in PBST. The membrane was incubated overnight at 4 °C with primary antibody dilutions in 5% skim milk in PBST according to Appendix A. The membrane was washed extensively with PBST the following day. Subsequently, the membranes were re-probed with appropriate HRP-conjugated secondary antibodies (refer to Appendix A) for 1 h at RT. The chemiluminescence HRP signal was detected with a SuperSignal™ West Festo Maximum Sensitivity Substrate kit (Thermo Fisher, Cat. #34094) using a ChemiDoc™ Touch Imaging System (Bio-Rad) equipped with ChemiDoc XRS software version 6.1 (Bio-Rad). The ratio of the optical density of the protein to the internal control (β-actin or HSP-90) was obtained and was expressed as a ratio or percentage of the control value in the figures.

## 5. Conclusions

In the present study, we combined multiplex peptide microarray technology and phosphoproteomics to identify novel targets downstream of cGMP/PKG signaling, and a selection of these were then confirmed using immune-based techniques. The integrated analysis revealed common pathways altered by PKG inhibition, notably, pathways related to the activity of the CREB transcription factor, as well as CaMK2 and CaMK4. Our results, therefore, provide new insights into cGMP-mediated retinal degenerative diseases and could be employed for the development of new disease biomarkers and therapies for their treatment.

## Figures and Tables

**Figure 1 ijms-25-03446-f001:**
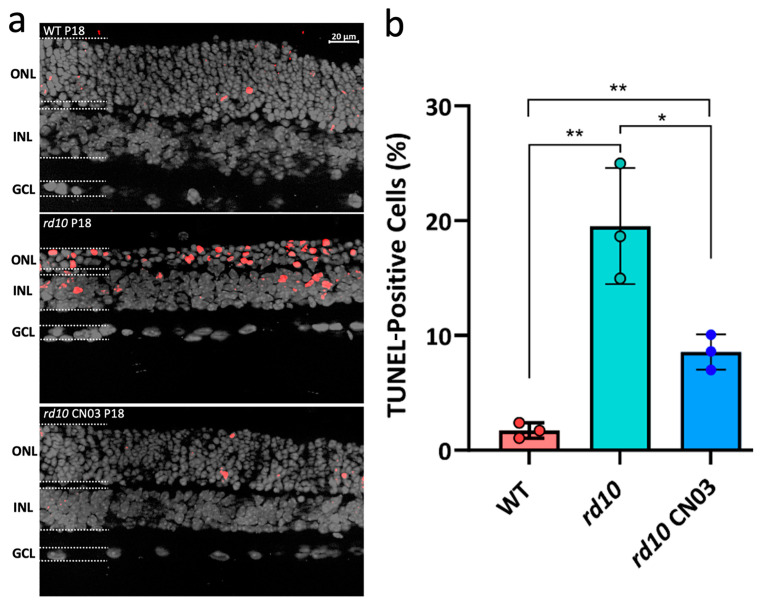
CN03-mediated photoreceptor protection in *rd10* retinal explant cultures. Retinal cross-sections of wild-type (WT) and *rd10* explant cultures. Retinas were explanted at post-natal day (P) 8 and cultivated with/without 50 µM of CN03 from P10 to P18. (**a**) The TUNEL assay (red) highlighted dying cells, especially in the outer nuclear layer (ONL). DAPI (grey) was used as a nuclear counterstain. INL = inner nuclear layer; GCL = ganglion cell layer. (**b**) Bar graph showing percentages of TUNEL-positive cells for WT, *rd10*-untreated, and CN03-treated *rd10* retinal explants. *n* = three different retinal explants per genotype/condition; error bars indicate standard deviation. Significance levels were determined via one-way ANOVA with Dunnett multiple comparisons; significance levels: * (*p* ≤ 0.05), ** (*p* ≤ 0.01).

**Figure 2 ijms-25-03446-f002:**
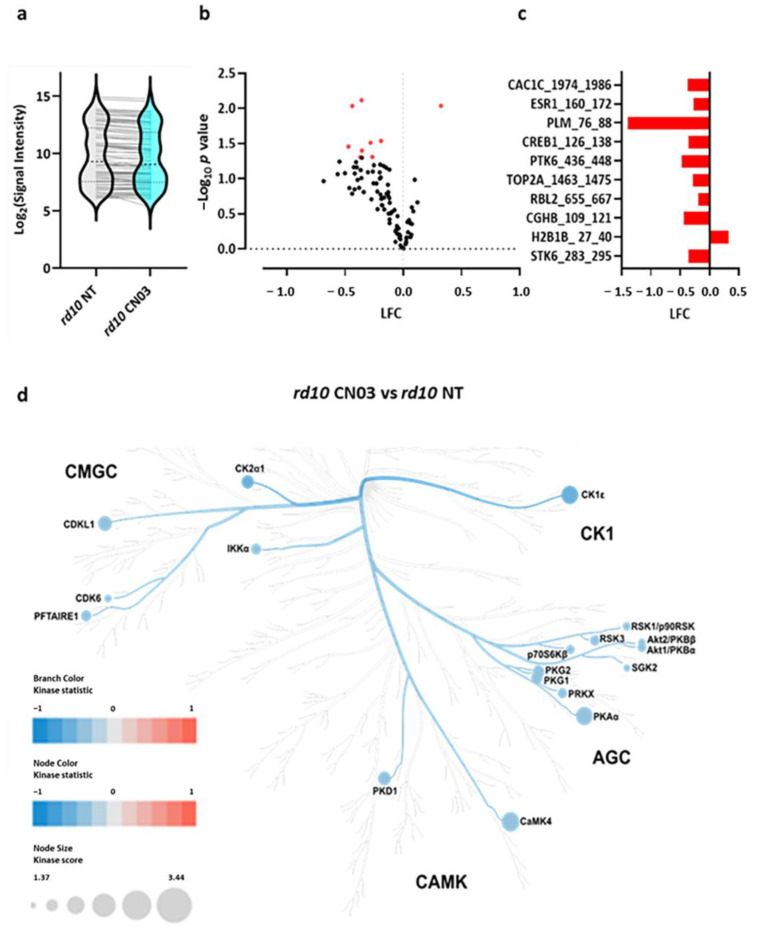
Serine/threonine kinase (STK) activity in *rd10* retinal explants after PKG inhibition. Retinal explant cultures were either non-treated (NT) or treated with 50 µM of CN03 (*rd10* NT, *n * =  10; *rd10* CN03, *n*  =  10). The kinase activity of retinal lysates was measured on PamChip^®^ Serine/Threonine Kinase (STK) arrays. (**a**) Violin plot representing the phosphorylation signal intensity of all the peptides on PamChip^®^ STK arrays as a Log2 signal intensity and their intensity value distribution when comparing *rd10* to *rd10* CN03-treated retinal explants. (**b**) Volcano plot showing the Log Fold Change (LFC) and −Log_10_
*p* value of the phosphorylated peptides. Red dots represent peptides with significant differences in phosphorylation (*p* < 0.05) as assessed by a paired *t*-test. Black dots represent peptides with no significant phosphorylation change. (**c**) Peptides displaying significant changes in phosphorylation are shown as a bar plot with their respective LFC. (**d**) The predicted kinases whose activities were most likely to be altered by the CN03 treatment (see Table 1) are visualized as a kinome phylogenetic tree where the branch and node color are encoded by the kinase statistic, with values < 0 (in blue) representing decreased kinase activity in *rd10* CN03-treated retinal explants. The node size is indicated by the kinase score, which is based on the specificity and selectivity of that kinase for a particular peptide.

**Figure 3 ijms-25-03446-f003:**
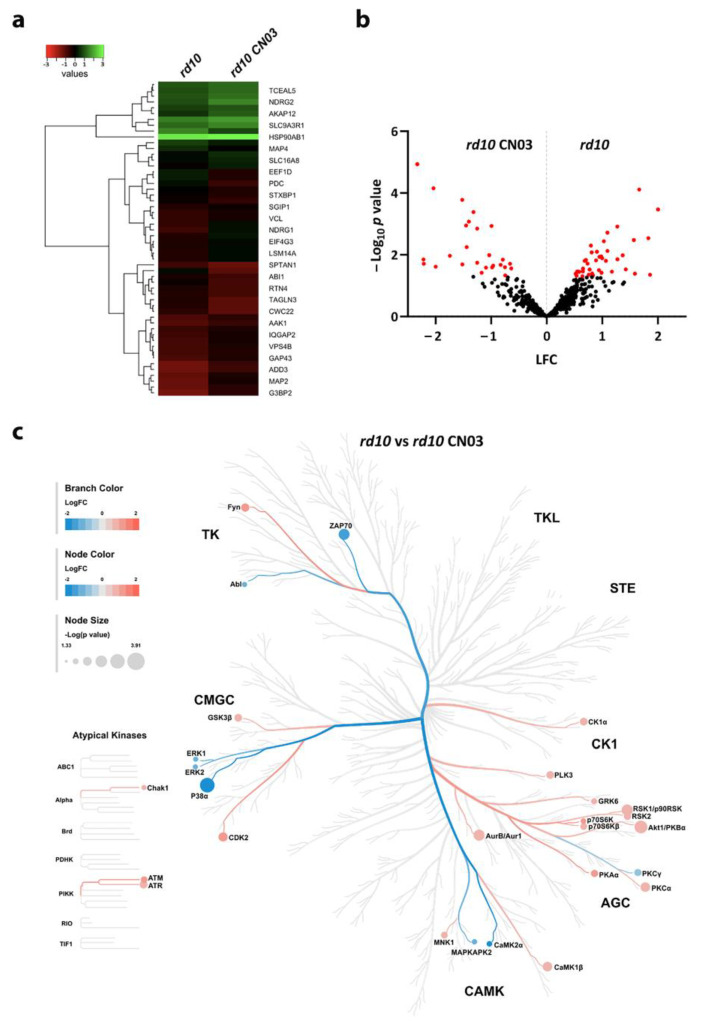
Phosphorylated sites in *rd10* retinal explants in response to PKG inhibition. Retinal explant cultures were treated or not with 50 µM of CN03, followed by phosphopeptide enrichment and mass spectrometry analysis. (**a**) Heatmap representing proteins with significantly different phosphorylations as their Log_2_ signal intensity. A red color indicates the relatively lower Log_2_ signal intensity of the protein, while a green color indicates one that is relatively higher. (**b**) Volcano plot showing the Log Fold Change (LFC) and −Log_10_ *p*-value of the phosphorylated sites. Red dots represent peptides/sites with significant changes in phosphorylation (*p* < 0.05), black dots indicate peptides with no significant phosphorylation change. (**c**) Kinases predicted to have significantly altered activity are visualized as a kinome phylogenetic tree, with their branch and node color encoded by the kinase statistic and values < 0 (in blue) and >0 (in red) representing the kinase activity as decreased or increased, respectively, in *rd10* CN03 retinal explants. The node size is indicated by the kinase score which is based on the specificity and selectivity of that kinase for a particular peptide.

**Figure 4 ijms-25-03446-f004:**
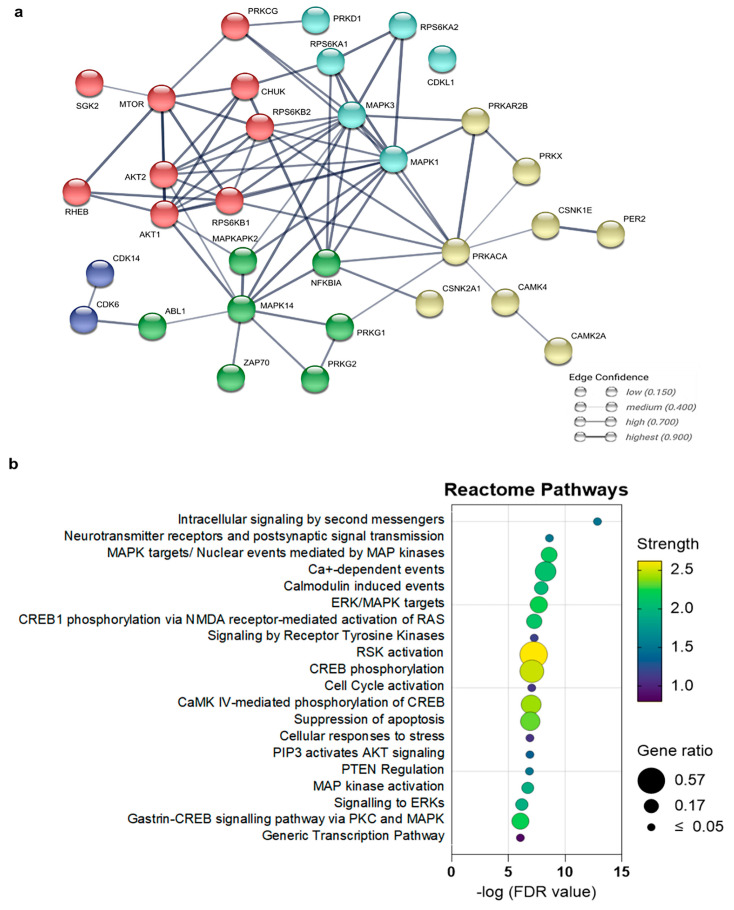
Pathway and network analysis for integrated kinome activity and phosphoproteomic data of CN03-treated *rd10* retinal explants. (**a**) Network of kinases with potentially altered activity in CN03-treated *rd10* retina. (**b**) Key biological pathways with potentially altered activity in the CN03-treated *rd10* situation. The false discovery rate (FDR) is the *p*-value adjusted for multiple tests using the Benjamini–Hochberg procedure, measuring the significance of enrichment. Pathways are ranked according to their FDR values and colored by their strength. Strength describes how large the enrichment effect is and is calculated as Log_10_ (no. observed proteins/no. of expected proteins). The node size indicates the gene ratio, i.e., the percentage of total genes or proteins in the given Reactome pathways (only input genes or proteins with at least one Reactome pathway were included in the calculation).

**Figure 5 ijms-25-03446-f005:**
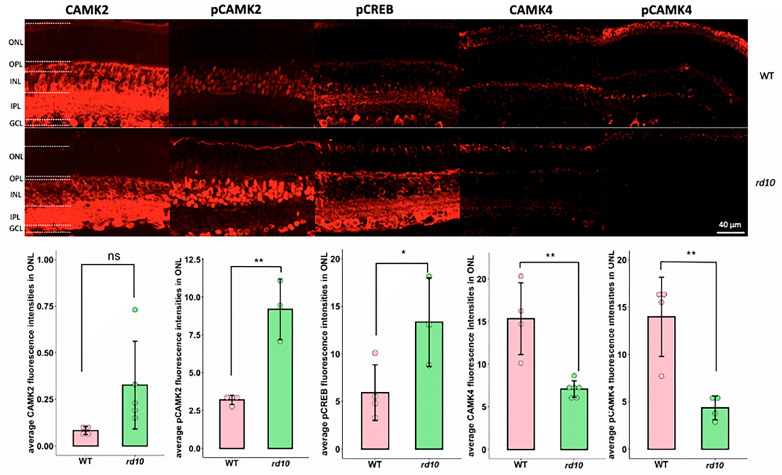
Presence and localization of PKG target proteins in the WT and *rd10* retinas. Immunostaining for Ca^2+^/calmodulin-dependent kinases (CaMK) and phosphorylated cyclic-AMP-response-element-binding (pCREB). The micrograph panels show retinal cross-sections derived from wild-type (WT; top row) and *rd10* (bottom row) P18 mice and stained with antibodies for CAMK2, pCAMK2, CAMK4, pCaMK4, or pCREB. Bar diagrams under the micrographs indicate the fluorescence intensities of the respective antibody staining, which were calculated as arbitrary fluorescence units/µm^2^ in the indicated area of interest (ONL). The fluorescence was adjusted with overexposure in the images for presentation purposes. OPL = outer plexiform layer, INL = inner nuclear layer, IPL = inner plexiform layer, GCL = ganglion cell layer. Significance levels: * (*p* ≤ 0.05), ** (*p* ≤ 0.01), ns = not significant.

**Figure 6 ijms-25-03446-f006:**
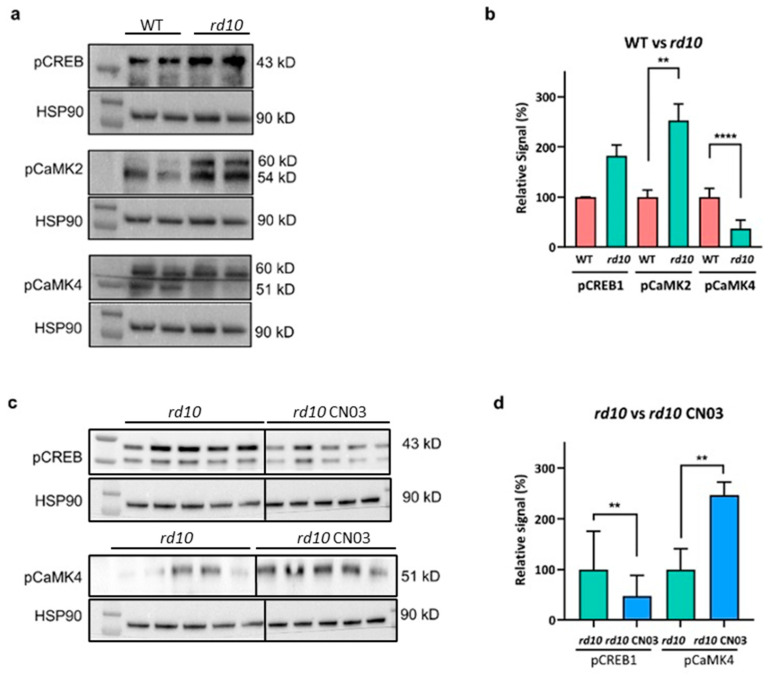
Confirmation of PKG target proteins in retina by Western blot. (**a**) Representative Western blots showing protein levels for the PKG targets pCREB (43 kD), pCaMK2 (54 kD), and pCaMK4 (51 kD) in the retina. First row of each blot is a marker lane. (**b**) Quantitative analysis with *n* ≥ 3 retinal tissues for each group. (**c**) Protein levels for PKG targets pCREB and pCaMK4 in retinal explants treated or not with CN03 (50 µM). First row of each blot is a marker lane. (**d**) Quantitative analysis of ≥ 5 retinal explants for each group. The significant differences between the groups were determined via unpaired Student’s *t*-test with significance indicated as ** (*p* ≤ 0.01) and **** (*p* ≤ 0.0001).

**Table 1 ijms-25-03446-t001:** Upstream kinase analysis results for *rd10* vs. *rd10* CN03 based on chip data. The kinase score includes the sum of the significance and specificity scores (cut-off kinase score ≥ 1.2 shown in the table). The kinase statistic shows the overall change of the peptide set that represents a given kinase. The negative values indicate lower activity in *rd10* CN03. For details, refer to Section 4.6, ‘Kinase activity measurements by kinome array’.

Rank	Kinase Name	Kinase Uniprot ID	Kinase Score	Kinase Statistic
1	CaMK4	Q16566	3.44	−0.43
2	PKAα	P17612	3.26	−0.31
3	CK1ε	P49674	3.24	−0.52
4	CDKL1	Q00532	2.72	−0.42
5	PKD1	Q15139	2.52	−0.42
6	PKG2	Q13237	2.32	−0.31
7	CK2α	P68400	2.39	−0.47
8	PKG1	Q13976	2.27	−0.31
9	PFTAIRE1	O94921	2.04	−0.44
10	IKKα	O15111	2.00	−0.43
11	PRKX	P51817	1.87	−0.30
12	RSK1/p90RSK	Q15349	1.94	−0.39
13	Akt1/PKBα	P31749	1.72	−0.30
14	P70S6Kβ	Q9UBS0	1.72	−0.31
15	RSK3	Q15418	1.58	−0.34
16	mTOR/FRAP	P42345	1.60	−0.37
17	CDK6	Q00534	1.47	−0.35

**Table 2 ijms-25-03446-t002:** *rd10* retinal proteins showing reduced phosphorylation after CN03 treatment. The table shows proteins with decreased phosphorylation from the mouse *rd10* explants after CN03 treatment. Retinal explant cultures were either treated with 50 µM of CN03 or left untreated, followed by phosphorylated peptide enrichment and MS analysis. For the selection of proteins, refer to Section 4.8; MS acquisition and analysis. Note that a given protein may appear more than once if it has more than one peptide affected, and that a given peptide may have more than one phosphorylation site. The total number of proteins in the list is 21.

S. No	Uniprot ID	Gene Symbol	Protein Description	Position	Fold Change	*p*-Value
1	P27816	MAP4	Microtubule-associated protein 4	352S, 354T	−2.33	1.17 × 10^−5^
2	Q9UQ35	SRRM2	Serine/arginine repetitive matrix 2	1097S	−2.04	7.04 × 10^−5^
3	Q9UI15	TAGLN3	Transgelin 3	163S	−1.52	1.68 × 10^−4^
4	P17600-1	SYN1	Synapsin I	568S	−1.32	4.17 × 10^−4^
5	Q9UJU6	DBNL	Drebrin like	278T	−1.40	8.52 × 10^−4^
6	Q9UJU6	DBNL	Drebrin like	277S	−1.45	1.13 × 10^−3^
7	Q13813	SPTAN1	Spectrin alpha, non-erythrocytic 1	1217S	−0.99	1.16 × 10^−3^
8	Q9HCG8	CWC22	CWC22 spliceosome-associated protein homolog	903S	−1.25	1.43 × 10^−3^
9	Q13427	PPIG	Peptidylprolyl isomerase G	685S	−1.44	5.69 × 10^−3^
10	P27816	MAP4	Microtubule-associated protein 4	841S, 688S	−1.03	1.03 × 10^−2^
11	P20941	PDC	Phosducin	54S	−1.74	1.08 × 10^−2^
12	P46821	MAP1B	Microtubule-associated protein 1B	2252S	−2.22	1.42 × 10^−2^
13	Q8WY54	PPM1E	Protein phosphatase, Mg^2+^/Mn^2+^-dependent 1E	545S	−0.79	1.43 × 10^−2^
14	Q9P2E9	RRBP1	Ribosome-binding protein 1	786S	−1.25	1.80 × 10^−2^
15	P13637	ATP1A3	ATPase Na^+^/K^+^-transporting subunit alpha 3	10S	−2.21	1.96 × 10^−2^
16	Q9H3Q1	CDC42EP4	CDC42 effector protein 4	64S	−0.66	1.97 × 10^−2^
17	P29692	EEF1D	Eukaryotic translation elongation factor 1 delta	133S	−1.52	2.06 × 10^−2^
18	O00499	BIN1	Bridging integrator 1	332S	−0.83	2.14 × 10^−2^
19	Q9NQC3	RTN4	Reticulon 4	344S	−0.96	2.26 × 10^−2^
20	Q92597	NDRG1	N-myc downstream regulated 1	356T	−2.00	2.44 × 10^−2^
21	Q15773	MLF2	Myeloid leukemia factor 2	237S	−0.75	2.53 × 10^−2^
22	Q8IZP0	ABI1	Abl interactor 1	183S	−0.98	2.55 × 10^−2^
23	P46821	MAP1B	Microtubule-associated protein 1B	2068S	−1.09	2.62 × 10^−2^
24	P61764	STXBP1	Syntaxin-binding protein 1	516S	−0.64	2.76 × 10^−2^
25	Q9UQ35	SRRM2	Serine/arginine repetitive matrix 2	1077S	−1.17	3.85 × 10^−2^
26	P46821	MAP1B	Microtubule-associated protein 1B	1775S	−0.75	4.65 × 10^−2^

**Table 3 ijms-25-03446-t003:** The table shows proteins with increased phosphorylation in *rd10* explants after CN03 treatment. Retinal explant cultures were either treated with 50 µM of CN03 or left untreated, followed by phosphorylated peptide enrichment and mass spectrometry (MS) analysis. For the selection of proteins, refer to Section 4.8 ‘MS Acquisition and Analysis’. Note that a given protein may appear more than once if it has more than one peptide affected, and that a given peptide may have more than one phosphorylation site. The total number of proteins in the list is 36.

S. No	Uniprot ID	Gene Symbol	Protein Description	Position	Fold Change	*p*-Value
1	Q7Z4V5	HDGFL2	HDGF-like 2	635S	1.67	7.74 × 10^−5^
2	P11137	MAP2	Microtubule-associated protein 2	1654S, 1650T	2.00	3.42 × 10^−4^
3	P46821	MAP1B	Microtubule-associated protein 1B	1934Y	1.27	1.23 × 10^−3^
4	Q9UEY8	ADD3	Adducin 3	679S, 681S, 683S	1.09	1.91 × 10^−3^
5	Q9UN86-2	G3BP2	G3BP stress granule assembly factor 2	227T	1.83	2.92 × 10^−3^
6	Q9ULU8	CADPS	Calcium-dependent secretion activator	88S, 89S	1.57	3.33 × 10^−3^
7	Q9UDY2	TJP2	Tight junction protein 2	1136S	1.03	3.65 × 10^−3^
8	Q8N111	CEND1	Cell cycle exit and neuronal differentiation 1	9S, 10S	0.80	5.06 × 10^−3^
9	O75351	VPS4B	Vacuolar protein sorting 4 homolog B	102S	1.09	7.59 × 10^−3^
10	Q02952	AKAP12	A-kinase anchoring protein 12	584S, 583T	0.90	8.04 × 10^−3^
11	Q08J23	NSUN2	NOP2/Sun RNA methyltransferase 2	723S, 717T	0.81	8.40 × 10^−3^
12	Q9NXV6	CDKN2AIP	CDKN2A interacting protein	168S, 169S	1.37	1.04 × 10^−2^
13	Q13576	IQGAP2	IQ motif containing GTPase activating protein 2	16S	0.98	1.13 × 10^−2^
14	Q5VTR2	RNF20	Ring finger protein 20	136S, 138S	0.95	1.17 × 10^−2^
15	P17677	GAP43	Growth associated protein 43	103S	1.00	1.30 × 10^−2^
16	Q9UN36	NDRG2	NDRG family member 2	332S, 330T	1.27	1.40 × 10^−2^
17	Q2M218	AAK1	AP2 associated kinase 1	936S	0.71	1.47 × 10^−2^
18	Q9Y4F1	FARP1	FERM, ARH/RhoGEF and pleckstrin domain protein 1	373S	0.89	1.53 × 10^−2^
19	Q5H9L2	TCEAL5	Transcription elongation factor A like 5	120S, 124S, 117T	0.69	1.58 × 10^−2^
20	Q8ND56	LSM14A	LSM14A mRNA processing body assembly factor	182S, 183S	1.07	1.59 × 10^−2^
21	O00567	NOP56	NOP56 ribonucleoprotein	543S	0.74	1.95 × 10^−2^
22	P09651	HNRNPA1	Heterogeneous nuclear ribonucleoprotein A1	4S, 6S	0.65	2.82 × 10^−2^
23	Q9BVG4	PBDC1	Polysaccharide biosynthesis domain containing 1	184S	1.43	2.96 × 10^−2^
24	O95907	SLC16A8	Solute carrier family 16 member 8	422S, 428S	0.77	3.00 × 10^−2^
25	Q9UQ35	SRRM2	Serine/arginine repetitive matrix 2	454S	0.94	3.13 × 10^−2^
26	O14745	SLC9A3R1	SLC9A3 regulator 1	283S, 285S, 286S, 288T	0.65	3.47 × 10^−2^
27	P46821	MAP1B	Microtubule-associated protein 1B	1792Y	0.55	3.48 × 10^−2^
28	P08238	HSP90AB1	Heat shock protein 90 alpha family class B member 1	255S	0.57	3.50 × 10^−2^
29	Q04637	EIF4G1	Eukaryotic translation initiation factor 4 gamma 1	1597S	1.17	3.52 × 10^−2^
30	Q9H6Z4	RANBP3	RAN-binding protein 3	57S, 58S	0.58	3.57 × 10^−2^
31	Q8ND76	CCNY	Cyclin Y	324S, 326S	0.99	3.83 × 10^−2^
32	Q9BQ15	SGIP1	SH3GL interacting endocytic adaptor 1	409T	0.52	4.07 × 10^−2^
33	O43432	EIF4G3	Eukaryotic translation initiation factor 4 gamma 3	267S	0.82	4.08 × 10^−2^
34	Q7Z4V5	HDGFL2	HDGF-like 2	366S, 367S	1.59	4.09 × 10^−2^
35	P46821	MAP1B	Microtubule-associated protein 1B	1788S, 1789S, 1793S	0.53	4.25 × 10^−2^
36	P21964	COMT	Catechol-O-methyltransferase	260S, 261S	0.76	4.42 × 10^−2^
37	Q92597	NDRG1	N-myc downstream regulated 1	366T	1.86	4.45 × 10^−2^
38	Q9UQ35	SRRM2	Serine/arginine repetitive matrix 2	962S, 964S, 963T	0.83	4.68 × 10^−2^
39	P18206	VCL	Vinculin	290S	0.54	4.80 × 10^−2^
40	P35579	MYH9	Myosin heavy chain 9	1943S	0.64	5.00 × 10^−2^

## Data Availability

The datasets used and/or analyzed in the current study are available from the corresponding authors on reasonable request.

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
