# Peer review of "Integrative Kinase Activity Profiling and Phosphoproteomics of rd10 Mouse Retina during cGMP-Dependent Retinal Degeneration"

_ijms, 2024, doi:10.3390/ijms25063446_

Round 1

Reviewer 1 Report

Comments and Suggestions for Authors

In this study, multiplex peptide microarray technology and phosphoproteomics were combined to identify novel cGMP/PKG signaling targets, and the selection of 666 of them was further confirmed by immune-based techniques. Integrated analytical analysis revealed common pathways altered by PKG inhibition, particularly pathways related to the activity of the transcription factor CREB1, as well as CaMK2 and CaMK4. These results provide new insight into cGMP-dependent retinal degenerative diseases.

The authors address the very important issue of cGMP/PKG-dependent photoreceptor degeneration and suggest important biomarkers for future PKG-based clinical therapies. The methods used are well described and the results are well written and well-illustrated. This makes it easier to keep track of the paper.

Especially the discussion part is interesting and the most important literature is cited. This makes the paper very attractive for the readers of International Journal of Molecular Sciences. 

Author Response

We thank you for your kind review of our manuscript.

Reviewer 2 Report

Comments and Suggestions for Authors
  1. The study could be susceptible to bias introduced during the phosphoproteomic profiling process. Factors such as sample preparation methods, variations in experimental conditions, and data analysis techniques may influence the identification and quantification of phosphorylation events. It is essential for the authors to address any potential sources of bias and validate the reliability of their phosphoproteomic data.
  2. The validation of PKG targets like CREB1 and CaMKs using immunostaining and immunoblotting is valuable. However, the authors should acknowledge the limitations of these techniques, such as the potential for nonspecific binding or antibody cross-reactivity. Additionally, alternative validation methods, such as mass spectrometry-based assays or functional assays, could provide further confirmation of the identified PKG substrates.
  3. While the study focuses on the rd10 mouse model for IRD, it's important to recognize that findings from animal models may not always directly translate to human diseases. The authors should discuss the relevance of their findings to human IRDs and consider the potential limitations of extrapolating results from animal models to clinical settings.
  4. The study identifies both known and novel PKG substrates involved in photoreceptor degeneration. However, the functional significance of these substrates and their contribution to the pathophysiology of IRDs require further investigation. The authors should provide a balanced interpretation of their results, considering alternative explanations and potential confounding factors.

Comments on the Quality of English Language

nil

Round 2

Reviewer 2 Report

Comments and Suggestions for Authors

The abbreviation cGMP in the title needs to be written in full form. 

All other concerns raised in the first round of review were addressed by the reviewers.